



# Localised geomorphic response to channel-spanning leaky wooden dams

Joshua M. Wolstenholme[1,2], Christopher J. Skinner[3,2], David Milan[4], Robert E. Thomas[2] and Daniel R. Parsons[1]

[1]Geography and Environment, Loughborough University, Loughborough, UK
[2]Energy and Environment Institute, University of Hull, Hull, UK
[3]FloodSkinner, UK
[4]School of Environmental Sciences, University of Hull, Hull, UK

*Correspondence to*: Joshua M. Wolstenholme (j.m.wolstenholme@lboro.ac.uk)

**Abstract.** The introduction of leaky wooden dams (or engineered log jams) into river corridors in low order steams in upper catchments has recently become a popular form of natural flood management, particularly in NW Europe. Leaky wooden dams are designed to emulate processes such as those of naturally occurring large wood in river systems, aiming to reduce downstream flood risk through the attenuation of water during higher flows, decreasing in-channel velocities and increasing channel-floodplain connectivity. Leaky wooden dams effectively act as channel roughness agents that disrupt the fluvial and

hydrological regime and attenuate the peaks in high river flows thus mitigating downstream flood risk. Despite their widespread installation, there is a paucity of data and understanding concerning the longer-term fluvial geomorphological response to leaky wooden dam installation. Here we present detailed quantification of both the geomorphic and sedimentary response to the installation of two leaky wooden dams in a catchment in Dalby Forest (North Yorkshire, UK) using high-resolution terrestrial laser scanning and detailed bathymetric surveys over a 2.5-year period. This period included two major

storms with a recurrence interval of 3.9 years and 3.4 years, and a further four smaller storm events (1.22–2.3 years). Results show that when leaky wooden dams are engaged by the river flow, local topographic complexity significantly increases as sediment transport pathways are perturbed. The flow field complexity additionally changes channel bed grain-size distribution with trends of fining upstream and coarsening downstream of the structure observed. The leaky wooden dam was also observed to generate scour pools downstream of the structure, and coarsen the armour layer through

winnowing of fines. Monthly observations revealed that channel topography and bed sediment patterns self-organise in response to sustained low flows and are perturbed by higher flow events. The findings highlight how frequent monitoring of different leaky wooden dam designs and structures under various flow conditions is vital to understand their longer-term impacts. Moreover it is critical that such observations are extended over longer-term periods in order to fully assess the efficacy of the structures as the channels respond to installations and the evolution of the geomorphic response. Finally,

additional work is also required to better consider how individual leaky wooden dams influence local geomorphology and alter sediment transport connectivity throughout the catchment.



## 1. Introduction

Large wood, commonly defined as >0.1 m in diameter and >1 m in length (Comiti et al., 2016), is used globally to manage flood risk through reducing stream velocities (Abbe and Montgomery, 2003; Grabowski et al., 2019; Lo et al., 2021; Wohl, 2015; Wohl et al., 2016; Wohl and Scott, 2017); reintroduction of wood to the river channel is a popular form of natural flood management (NFM), accounting for approximately 20% of UK projects focused on habitat creation, river engineering, and downstream flood hazard reduction (Cashman et al., 2018; Grabowski et al., 2019; Gurnell, 2012; Ruiz-Villanueva et al., 2019; Wohl, 2019). Large wood can drastically increase fluvial complexity (Wohl et al., 2019) through altering the local hydraulic regime whilst being resistant to erosion and providing storage space for water through decreasing longitudinal connectivity (Gurnell et al., 2018). Large wood can also temporarily store water both within the channel (online) and on the floodplain (offline), and it is through this retardation of the flow that wood can provide a suppression of flood peaks (Dadson et al., 2017; Gurnell et al., 2018). Natural riverine large wood in industrialised countries, especially Britain, is relatively rare, especially in channels wider than 10 m, and others have shown that it is important to emulate and support the recruitment of natural wood where appropriate (Gurnell et al., 2018, 2002). Therefore instream structures are now often constructed to emulate processes caused by natural wood, and to aid wood recruitment when there is a lack of natural large wood in the system (Addy and Wilkinson, 2016; Dixon et al., 2018).

Large wood causes significant alterations to hydraulics through increasing roughness and obstructing flow (Gippel, 1995), reducing stream velocities (Gallisdorfer et al., 2014; Schalko et al., 2021) and sediment transport both in suspension (Parker et al., 2017; Walsh et al., 2020) and as bedload (Clark et al., 2022; Spreitzer et al., 2021). The formation of underflow and plunge pools is often observed both immediately downstream of natural and introduced large wood (Buffington et al., 2002; Larson et al., 2001; Montgomery et al., 1995; Wohl and Scott, 2017), increasing geomorphic heterogeneity and providing more diverse habitats (Hafs et al., 2014; Klaar et al., 2011). Large wood can also both induce and limit bank erosion depending on specific interactions, particularly where flow is concentrated to one side of the channel (Buffington et al., 2002). Lateral floodplain connectivity can also be increased by the presence of large wood and its mobility, facilitating the transition from single-thread channels to anastomosing or anabranching systems (Bertoldi et al., 2014; Wohl, 2013), which can lead to improved ecological conditions (Lo et al., 2024; Roni et al., 2015; Wohl, 2019).

Leaky wooden dams (or engineered log jams, herein referred to as LDs) are a type of NFM intervention that are often installed in sequences in upper catchments in permanent or ephemeral streams (Wren et al., 2022) to emulate the potential benefits of large wood. The overarching aim of LDs is to store greater volumes of water in the upper catchment and then slowly release that water over time, extending and flattening the downstream flood peak through increasing flooding locally in the immediate area around the LDs (Roberts et al., 2023). LDs effectively increase channel roughness and reduce flow conveyance, while increasing floodplain connectivity and decreasing longitudinal connectivity, similar to natural wood (Gippel, 1995; Roberts et al., 2023; Wohl and Beckman, 2014; Wren et al., 2022). LDs are often installed with a vertical gap at the river bed to allow base level flows to pass unimpeded, enabling the movement of wildlife and reducing the likelihood



of blockages (Dodd et al., 2016). The presence of a gap also increases flow and habitat diversity, whilst increasing the potential for geomorphic heterogeneity through the development of pool and riffle sequences, as well as sediment storage upstream of the LD (Follett et al., 2021; Lo et al., 2022; Wohl et al., 2016), and overall sediment dis-connectivity (Burgess-Gamble et al., 2017; Grabowski et al., 2019; Poeppl et al., 2023; Wenzel et al., 2014; Wren et al., 2022).

Despite a range of previous research concerned with the impacts of large wood on river systems (e.g., Abbe and
Montgomery, 2003; Wohl et al., 2019), there has been little field research into the influence of channel spanning LDs on local geomorphology, with only one field-based study specifically focusing on LDs and their impacts. Lo et al. (2022) used topographic and bathymetric observations to assess the impact of the LDs on bank erosion, sediment storage, pool formation and LD instability. Although grain-size distribution (herein GSD) was not reported, the authors stated that $D_{50}$ was in the cobble range (64–256 mm), but not how this evolved over time. Despite the popularity installing LDs for flood risk
management across the UK, the main limitation of this research (as identified by the authors) is the lack of studies in different contexts (geology, catchment morphometries, drainage densities and land use for example). There is thus a knowledge gap that requires a more robust evidence base, including evaluations of other LD designs in different geological and climate regions, particularly in regard to understanding longer term impacts and catchment scale responses to installations of LDs over a range of time and space scales.

A key unknown of is how the GSD around LD structures evolves through time, and potential controls this may have on channel roughness. Understanding how the GSD evolves is an important component for numerical modelling; Skinner et al. (2018) performed a global sensitivity analysis of the CAESAR-Lisflood using the Morris (1991) method, and found that GSD is the fourth most influential parameter influencing sediment efflux (out of 15), where the top three were modelling parameters including the choice of sediment transport law, slope for edge cells and vegetation critical shear stress.
Additionally, Durafour et al., (2015 cited by Lepesqueur et al., 2019) identified that modelling with a uniform grain size can lead to an overprediction of fluvial sediment flux, therefore GSD is a critical component of numerical modelling, and it is imperative that it is correctly quantified and implemented.

In the present paper, we explore the influence of two LD designs on local geomorphological changes over time and build on recent research (Lo et al., 2022) using terrestrial laser scanning and bathymetric surveys to monitor topographic change.
Monitoring occurred for 2.5 years and included two storms that had an estimated recurrence interval (herein RI) of 3.9 and 3.4 years, and a further four events where RI was 1.22–2.3. We additionally examine and monitor changes to GSDs around installed LDs. The objectives of this paper were to understand the influence of two different LD structures on the directionality and magnitude of geomorphic change around a single structure (herein referred to as the unit-scale). Furthermore, we aim to highlight geomorphic variability induced by the structures, as well as the importance of frequent
monitoring to accurately assess long-term impacts at the local, unit-scale.



## 2. Methods

### 2.1. Study site

Dalby Forest is a commercial woodland located within the North York Moors National Park, UK, maintained by Forestry England. In January 2020, 14 in-channel LDs were installed in Staindale Beck as part of the Derwent Rivers NFM
demonstration project (Lavelle et al., 2019). Staindale Beck is a second order gravel-bed stream with a gradient of 0.011 m m$^{-1}$ that drains a 12 km$^2$ catchment with elevation ranging from 107–240 mAOD (metres Above Ordnance Datum Fig. 1b). Upstream geology is composed of a mixture of mudstones and sandstones, with some Holocene alluvium (clays, silts, sands and gravels).

The catchment is characterised by woodland (70%), with grassland (17%) and heather (10%) the other principle land use
types (Marston et al., 2022) as shown in Fig. 1a. It has a mean annual precipitation of 980 mm with monthly averages ranging from 55.6–118.9 mm (1991–2020 mean, Fylingdales weather station at 262 mAOD, 11.2 km NE; Met Office, 2020). The catchment contains 60.2 km of river corridor, with a mixture of first (9.9 km), second (3.6 km) and third-order (3.5 km) streams following the Strahler (1957) stream order notation (Fig. 1a, b). The channels are between 1 and 4 m wide, widening towards the catchment outlet.





**Figure 1:** Location of studied leaky wooden dams in relation to the Dalby Forest catchment. (a) 2021 Land cover map with percentage cover shown in the legend (Marston et al., 2022). (b) Stream network generated from terrain data showing Strahler stream order and elevation (Ordnance Survey, 2020). Note same scale as (a). (c) Location of LD1 and LD2. (d) UK context for site location. (e) & (f) LD1 and LD2 respectively, flow direction denoted by white arrow. G) River level record 2008–2024 for Thornton Beck, Thornton Le Dale (Grid Reference: SE 83681 83418; Station ID: L2725) approximately 7 km downstream of study area (from Environment Agency) including estimated return period ($Q_n$) as described in the supplementary material.

### 2.1.1. Leaky dam structures

LDs were installed in late 2019 through live felling and anchoring using wooden stakes or embedding into a channel bank and being placed across the full width of the channel. Two of the 14 dams (referred to as LD1 and LD2 herein) were monitored since installation (including a pre-installation baseline survey undertaken in July 2019).



LD1 (Fig. 1e) is a 6.2 m wide channel spanning LD with a 0.28 m average vertical gap across its width at the time of installation. The channel was 2.7 m wide where the LD was installed. It was anchored *in situ* with a wooden stake on the true right bank to reduce the likelihood of displacement during high flows and was embedded into the surface on the true left bank. The structure was comprised of a single log which was fully separated from a tree trunk, with some thin branches attached facing upwards. As this LD is fully channel spanning and thus does not interact with baseflow. Thus, it is a "Type 1" LD (fully channel spanning, not interacting with baseflows) following the classification of Lo et al. (2021). The LD extends onto the floodplain by approximately 1.5 metres either side of the channel.

LD2 (Fig. 1f) was felled and left in situ, remaining attached to its stump to enable growth (a "living" LD). The LD itself was composed of two sub-sections, furthest downstream (LD2a) was an 8.55 m long section spanning a 3.85 m wide channel. The structure was partially submerged with a minimum gap height of 0.14 m, and a maximum of 0.43 m at the time of installation. LD2a was classified as a "Type 2" LD (fully channel spanning, interacting with baseflows but not touching the stream bed; Lo et al., 2021). LD2 was more complex than LD1 due to the presence of a pre-existing walkway that formed a secondary sub-structure LD2b, that was fixed with wooden anchors. LD2b was a "Type 1" LD (fully channel spanning, not interacting with baseflows at all; Lo et al., (2021). LD2b was installed above the average water depth, was 3.83 m wide, spanned a 3.25 m wide channel and had a 0.41 m gap. LD2a extended approximately 1.5 metres either side of the channel, whereas LD2b extended less than 0.5 m either side of the channel banks.

### 2.1.2. Hydrological summary

In lieu of long-term flow data, two Environment Agency-managed river level monitoring stations (L2725 & F25110) that have respectively been recording river level since February 2008 and July 2003 were used to perform flow frequency analysis to characterise the hydrological history of the catchment as described in the supplementary information and summarised here. L2725 is located approximately 7 km downstream of the study area on Thornton Beck in Thornton Le Dale (Fig. 1g) and F25110 in an adjacent catchment on Levisham Beck at Levisham Mill (see Supplementary Fig. 1).

Based on 21 years of stage data the maximum mean annual stage for the larger L2725 catchment was 0.358 m and 0.305 m for paired catchment F25110. Flow frequency analysis revealed the approximate Q2 flood requires stage to reach 0.343 m and 0.303 m for L2725 and F25110 respectively then 0.429 m and 0.377 m for RI=5 (see Supplementary Fig. 2). Due to uncertainties in data at F25110 when stage exceeds 0.4 m leading to out of bank flow (Hydrology NE Environment Agency, 2024), larger recurrence intervals are not reported. The flood of record differs for each catchment; L2725 achieved a level of 0.401 m (RI = 3.9) on June 12, 2020, following 23.6 mm of rain in 24 hours (captured at Brown Howe rain gauge [036225], 7.5 km NE of field site). The flood of record for F25110 on February 20, 2022, following named storms Dudley, Eunice and Franklin (February 16–21) peaked with a stage of 0.342 m (RI = 3.4). Other peaks in the flow data are reported in Table 1.





**Table 1:** Summary of key storm events recorded by L2725 and F25110. Flood of record for each gauge during the monitoring period denoted by †.

| Storm Name | Date of impact | L2725 | | F25110 | |
|---|---|---|---|---|---|
| | | Stage (m) | RI | Stage (m) | RI |
| Unnamed | 12/06/2020 | 0.401† | 3.90 | 0.111 | – |
| Francis | 25/08/2020 | 0.233 | 1.08 | 0.211 | 1.19 |
| Christoph | 19–22/01/2021 | 0.330 | 1.80 | 0.316 | 2.30 |
| Darcy | 06–08/02/2021 | 0.320 | 1.38 | 0.240 | 1.61 |
| Unnamed | 06/10/2021 | 0.191 | – | 0.209 | 1.22 |
| Unnamed | 04/11/2021 | 0.204 | – | 0.243 | 1.37 |
| Dudley, Eunice & Franklin | 16–21/02/2022 | 0.244 | 1.10 | 0.342† | 3.40 |

Further to individual storm events resulting in peak stage depths, the rainfall anomaly for winter of 2020/21 (12/2020–02/2021) was in excess of 170% compared to the 1991-2020 average for the study area and was one of the top ten wettest years on record (Kendon et al., 2022).

## 2.2. Water depth

Water depth was sampled at five-minute intervals using four absolute pressure transducers (Solinst Levelogger Edge M5) distributed approximately 5 m upstream and downstream of LD1 on suitable anchor points. Loggers were installed 5 m
downstream of LD2a with a second pressure transducer located between LD2a and LD2b due to the lack of suitable anchor points. Each pressure transducer was housed in a stilling well to dampen potential noise induced through waves. A Solinst Barologger was also installed to provide atmospheric compensation for the sensors. Data was downloaded approximately every three months to ensure a continuous record. The sensors were reinstalled after each download with manual measurements of the river depth taken to provide any necessary corrections for water surface elevations to be mapped to
depth and thus account for changes resultant from bedload transport, wood collapse or other disturbance. Individual data records were merged in MATLAB, and any gaps caused by the download period were filled using linear interpolation to ensure a continuous record. Noise not eliminated by the stilling well was reduced using a wavelet denoising algorithm (after Lockwood et al., 2022) using a fixed "*minimax*" threshold to minimize possible signal loss without removing peaks in the signal.

## 2.3. Grain-size distribution

The intermediate axis of, on average, 100 pebbles were measured at each ~3 month visit to provide an unbiased estimate of sediment distribution (Green, 2003; Wolman, 1954). A random walk approach was used within 5 m of the LD to collect two sample sets (upstream and downstream) at both sites. Measurements smaller than 0.5 mm were grouped into a single class in the field. $D_{16}$, $D_{50}$ and $D_{84}$ were extracted by creating a cumulative density function (CDF) where grain size is represented
by the average recorded value in each class (except for fine-grained measurements). Using GSDtools in RStudio (Eaton et





al., 2019), the estimated grain size and upper and lower uncertainty bounds for each of the grain size metrics were generated through equal area approximation of the binomial distribution. The final grain sizes were obtained through interpolation of the binned CDF data, generated by normalising the data relative to the number of records. The 95% confidence interval was used to calculate uncertainty.

## 2.4. Topography and bathymetry

Topography was captured using a terrestrial laser scanner (Topcon GLS-2000, herein referred to as TLS) referenced in a local coordinate system using a total station (Topcon OS-103, herein referred as TS) to ensure consistency between scans. At each LD, a minimum of two TLS scans were captured with a point density of 6.3 mm at 10 m, recording only the last return pulse to reduce noise caused by vegetation. To avoid obstructing objects and to reduce the potential of shadowing (Heritage and Hetherington, 2007), the TLS was placed on the channel bank approximately 5 m upstream and 10 m downstream of each LD. Scan positions were registered using a minimum of three target tie points distributed evenly throughout the reach, and then georeferenced to the local coordinate system in Topcon ScanMaster (v.3.0.7.4) to create a single referenced point cloud for each survey date. Bathymetry was captured using the TS through approximately one-metre point sampling broadly equally spaced but favouring breaks in slope and channel edges up- and downstream of the LDs, also in the local coordinate system (Heritage et al., 2009).

### 2.4.1. Data processing

Referenced point clouds were manually cropped to the area of interest. Vegetation was removed where possible to create a bare-ground point cloud for change analysis using the cloth simulation filter (CSF) plugin in CloudCompare (Zhang et al., 2016). The LD and any remaining channel overhanging vegetation were removed manually by adjusting the viewpoint of the point cloud and using the segment tool in CloudCompare by visually identifying vegetation based on prominence from the surrounding point cloud and site knowledge.

Finally, the processed point cloud was gridded in Surfer® 8. To generate the grid, minimum and maximum $x$ and $y$ coordinates were predefined to ensure an identical grid for all survey dates, with the same grid resolution, retaining the minimum elevation point in each cell to decrease the likelihood of including low-lying vegetation. A resolution of 0.1 m was chosen as this is coarser than the TLS and TS recorded accuracy (TOPCON, 2017a, b) and the horizontal georeferencing errors. The grid was generated using triangulation with linear interpolation which is an exact interpolator that preserves raw data points, as recommended by Schwendel et al. (2012).

### 2.4.2. Error quantification

TLS error was quantified through calculating tie point accuracy by comparing the TS coordinate data to the TLS tie point coordinates for each survey position. Residual distance (i.e., the distance between the measured point and the interpolated



point in 3D space; the raw elevation survey error) between tie points was calculated during registration for each dimension (i.e., $x$, $y$ and $z$). Across all point clouds, the georeferencing error was greater than internal registration error, as such the georeferencing error is used to assess the total error across an individual cloud. Horizontal error is typically within the same order of magnitude of the vertical error, but has little influence on vertical surface differences in most fluvial environments (Wheaton et al., 2010). Therefore, to assess registration quality, only the residuals in the $z$ dimension were considered by calculating the root mean square error (RMSE) as a measure of goodness of fit of the DEM to the surveyed points, commonly used to evaluate map accuracy (Fisher and Tate, 2006). To provide a more complete overview of error beyond RMSE, the mean error (ME), the standard deviation of the mean error (SD), and mean absolute error (MAE) were also calculated (Fisher and Tate, 2006; Schwendel et al., 2012).

All deviations shown in Table 2 are very low, generally sub-centimetre for both TS and TLS data. ME results are weakly positively biased, indicating the elevations were slightly overestimated during registration. SD and RMSE have a similar order of magnitude and are also consistent between surveys, for both TS and TLS, therefore these data are appropriate for further analysis (Schwendel et al., 2012). RMSE, SD and MAE for the TLS registration points are larger than those reported for the TS sample points, likely due to the influence of number of observations (five independent check points vs averages of 71 [site 1] and 110 [site 2] for TS). Nevertheless, all vertical errors for TS are <0.013 m indicating a high level of accuracy of the gridded DEM representing the bathymetric topography. All TLS vertical errors are <0.026 m, except for April and November of 2021 for LD1 which have a maximum error of 0.043 m and 0.034 m respectively.



**Table 2:** Error metrics for survey residuals. All values are reported in metres, the maximum error for each metric for each site is highlighted in bold.

| | | LD1 | | | | LD2 | | | |
|---|---|---|---|---|---|---|---|---|---|
| | | RMSE | ME | SD | MAE | RMSE | ME | SD | MAE |
| **TS** | 15/07/2019 | 0.006 | 0.006 | <0.001 | 0.003 | 0.005 | 0.004 | -0.001 | 0.003 |
| | 28/01/2020 | 0.005 | 0.005 | <0.001 | 0.003 | 0.007 | 0.007 | -0.001 | 0.004 |
| | 27/09/2020 | - | - | - | - | 0.010 | 0.010 | -0.001 | 0.004 |
| | 07/01/2021 | 0.007 | 0.007 | <0.001 | 0.004 | 0.007 | 0.007 | -0.001 | 0.004 |
| | 26/01/2021 | - | - | - | - | **0.013** | **0.013** | **-0.002** | **0.009** |
| | 07/04/2021 | **0.009** | **0.009** | <0.001 | **0.005** | 0.008 | 0.008 | -0.001 | 0.005 |
| | 20/08/2021 | 0.006 | 0.006 | <0.001 | 0.004 | 0.008 | 0.008 | <0.001 | 0.005 |
| | 30/11/2021 | 0.008 | 0.008 | **-0.002** | 0.004 | 0.008 | 0.008 | **-0.002** | 0.005 |
| | 11/02/2022 | - | - | - | - | 0.006 | 0.006 | <0.001 | 0.005 |
| | 14/04/2022 | 0.006 | 0.006 | <0.001 | 0.004 | 0.009 | 0.009 | <0.001 | 0.004 |
| | 11/08/2022 | - | - | - | - | 0.007 | 0.007 | <0.001 | 0.005 |
| **TLS** | 15/07/2019 | 0.006 | <0.001 | 0.006 | 0.004 | 0.021 | <0.001 | 0.024 | 0.018 |
| | 28/01/2020 | 0.004 | <0.001 | 0.005 | 0.004 | **0.026** | <0.001 | **0.029** | **0.024** |
| | 27/09/2020 | 0.009 | 0.001 | 0.010 | 0.008 | 0.014 | **-0.003** | 0.016 | 0.012 |
| | 07/01/2021 | - | - | - | - | 0.025 | 0.001 | 0.028 | 0.023 |
| | 26/01/2021 | 0.011 | -0.001 | 0.013 | 0.010 | 0.010 | <0.001 | 0.011 | 0.009 |
| | 07/04/2021 | **0.043** | **0.002** | **0.050** | **0.039** | 0.007 | 0.001 | 0.007 | 0.006 |
| | 20/08/2021 | 0.008 | <0.001 | 0.009 | 0.007 | 0.004 | 0.001 | 0.005 | 0.003 |
| | 30/11/2021 | 0.034 | <0.001 | 0.038 | 0.026 | 0.017 | 0.001 | 0.019 | 0.016 |
| | 11/02/2022 | 0.003 | <0.001 | 0.004 | 0.003 | 0.004 | <0.001 | 0.005 | 0.004 |
| | 14/04/2022 | 0.006 | <0.001 | 0.006 | 0.004 | 0.021 | <0.001 | 0.024 | 0.018 |
| | 11/08/2022 | 0.004 | <0.001 | 0.005 | 0.004 | **0.026** | <0.001 | **0.029** | **0.024** |

## 2.5. Change analysis

DEMs of difference (DoDs) were created by subtracting a reference DEM, in this case the baseline survey in July 2019, from a subsequent DEM, creating a localised elevation change model. To identify significant change in the DoD, they can be thresholded to identify stable parts of the landscape and remove these from the change analysis without requiring full error propagation, producing a level of detection (LoD) where there is significant elevation change (Milan et al., 2011; Wheaton et al., 2010). Thresholding can be applied globally across the DEM or spatially quantified (Milan, 2012; Milan et al., 2011). The former is more aggressive and likely to remove areas that are unstable, while the latter requires a higher point survey density (Milan et al., 2011). Here, the global method was used due to low point density for TS data. To calculate the LoD, the critical threshold error, $U_{crit}$, was calculated through propagating survey errors by combining the RMSE of each DEM as shown in equation 1.

$$U_{crit} = t\sqrt{(\sigma_{DEM1})^2 + (\sigma_{DEM2})^2} \qquad (1)$$



where $\sigma_{DEM1}$ and $\sigma_{DEM2}$ are the standard deviation of the residuals and $t$ is the critical $t$ value for the chosen confidence level (Milan et al., 2011). The $t$ value was set to the 95% confidence limit where $t \geq 1.96$; $2\sigma$ (Brasington et al., 2003; Milan et al., 2011). Anderson (2019) identified that thresholding could introduce bias into net change estimates, therefore two data sets were generated for analysis: one that had been thresholded to evaluate significant erosion and deposition throughout
time and space, and another that had not been thresholded that was used to evaluate net change.

Visual inspection of the TLS point clouds following post-processing revealed that a substantial amount of low-lying vegetation remained after cleaning the point cloud that proved impractical to remove. Cloud-to-cloud analysis and DoDs were therefore inappropriate due to the volume of vegetation present for TLS derived data. To mitigate the potential loss of valuable information, planform change was calculated through extracting horizontal profiles at the same elevation for the
July 2019 and April 2022 surveys to identify areas of erosion relative to the LD.

Point clouds were detrended by fitting a plane and applying the inverse transformation using CloudCompare. The clouds were then filtered to a local elevation where there was minimum vegetation across both scans. Bank profiles were extracted manually—through point picking— terminating equidistant to one another perpendicular to flow. Bank profiles were then compared in MATLAB using Gauss's shoelace method (Braden, 1986). The shoelace method returns the area of a polygon
through calculating the total of matrix determinants of subsequent coordinates progressing in the same direction (clockwise or anti-clockwise), finishing at the start coordinates as shown in equation 2.

$$Area = \frac{1}{2}\left\{ \begin{bmatrix} x_0 & x_1 \\ y_0 & y_1 \end{bmatrix} + \begin{bmatrix} x_1 & x_2 \\ y_1 & y_2 \end{bmatrix} + \cdots + \begin{bmatrix} x_{n-2} & x_{n-1} \\ y_{n-2} & y_{n-1} \end{bmatrix} + \begin{bmatrix} x_{n-1} & x_0 \\ y_{n-1} & y_0 \end{bmatrix} \right\} \tag{2}$$

where $\begin{bmatrix} x_n & x_{n+1} \\ y_n & y_{n+1} \end{bmatrix}$ are coordinate pairs. Often the formula uses the absolute of the determinant of the coordinate pairs, however here the sign of the area indicates erosion (negative) and deposition (positive).

## 3. Results

### 3.1. Water depth

Average water depth at site 1 was 0.26 m and 0.21 m (up- and downstream respectively). Bankfull depth was 0.58 m, however, flow did not engage with LD1 at this site, nor did the channel exceed its banks during the monitoring period. Maximum depth was 0.54 m upstream of LD1 and 0.51 m downstream during Storm Christoph (21/01/2021; estimated RI of 2.3 years). In contrast site 2 average water depth was and 0.29 m and 0.17 m (up- and downstream respectively). Bankfull
depth was 0.44 m and was exceeded upstream for a total of 69 days (7.6%) of the monitoring period but not downstream. The true right bank became partially inundated and flow outflanked LD2b, resulting in increased scour and limiting the maximum capacity of the structure during these bankfull periods. The maximum recorded depth was 0.67 m upstream and 0.44 m downstream during Storm Darcy (07/02/2021), with an RI of 1.61 years. Seven storm events occurred during the





monitoring period, denoted by red arrows on Fig. 2 linked to key storms identified in Table 1, that resulted in near
exceedance or exceedance of bankfull stage at site 2, in addition to increased average water depth throughout winter 2020/21
observed at both sites.

Initially, water depths at LD1 covaried, averaging 0.21 m downstream and 0.22 m upstream between February 2020 and
November 2020, increasing to 0.31 m and 0.32 m, respectively, until February 2021, when the upstream depth diverged from
the downstream. Throughout the monitoring period, there was an average difference of 0.05 m and a maximum difference of
0.11 m between the two depths, with the upstream depth ranging from 0.20 m to 0.54 m, and the downstream depth ranging
from 0.13 m to 0.51 m.

In contrast, the upstream and downstream water depths quickly diverged at LD2, with an average difference of 0.12 m
between the upstream and downstream depths. Downstream peaks were generally shorter in duration than upstream (Fig.
2b). Additionally, once the LD was fully engaged with the river, there was a near constant difference between the
downstream and upstream depths of up to 0.27 m. This difference decreased to less than 0.10 m following the last notable
storm of the monitoring record in February 2022 (Dudley, Eunice and Franklin). Upstream and downstream depths ranged
from 0.13 m to 0.70 m and from 0.06 m to 0.44 m, respectively.

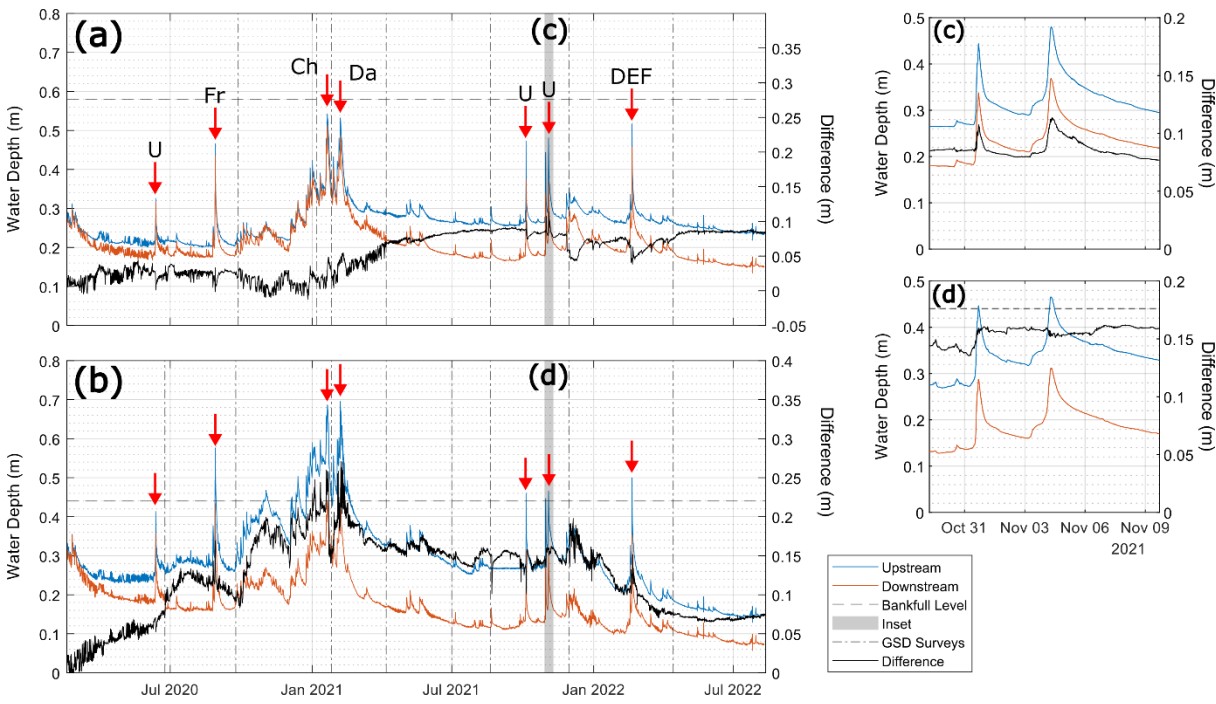

**Figure 2:** Temporal variations of water depth for LD1 (a) and LD2 (b) with difference between upstream and downstream recordings
shown, as well as key peaks and storms (U=unnamed, Fr=Francis, Ch=Christoph, Da=Darcy, DEF=Dudley, Eunice & Franklin). Also
shown are dates of GSD surveys and bankfull level. Note different y-axis scale. (c) and (d) show variations in water depth throughout time
between the two sites.





## 3.2. Grain-size distribution

When comparing the upstream and downstream GSDs, except for an initial increase of the sand and fine gravel fractions
upstream of the structure immediately following installation and until September 2020, the between-event variability is
greater than any consistent change to the GSD at LD1 during the monitoring period, although there is some evidence of
slight coarsening over the longer record (Fig. 3a, b). In contrast, the GSD at LD2 showed a consistent fining trend upstream
of the structure post-installation, resulting in a decrease of $D_{50}$ from 10 mm to < 1 mm (Fig. 3c) and a consistent coarsening
trend downstream of the structure, resulting in an increase of $D_{50}$ from 5 mm to 30 mm (Fig. 3d).

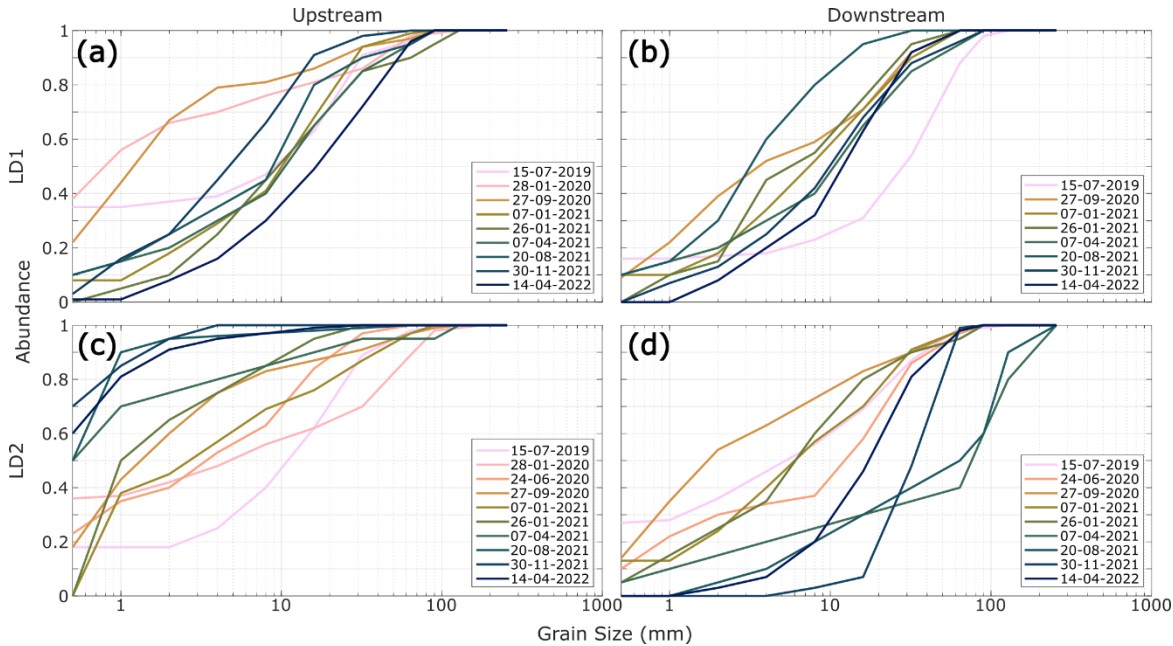


**Figure 3:** Cumulative density functions of Wolman pebble count data at LD1 (a) & (b) and LD2 (c) & (d). Sites are separated into
upstream (a) & (c) and downstream (b) & (d) observations.

$D_{50}$ and $D_{84}$ highlight the variability between the two sites as well as upstream and downstream of the LD structures as
shown in Fig. 4. LD1 showed little difference for all metrics, with upstream and downstream matching closely throughout
the monitoring period, often within the calculated 95% confidence interval of the measurements themselves, as such this
aligns with the more detailed GSD overview presented in Fig. 3. In contrast, metrics for LD2 diverge notably from January
2021 onwards following the wet 2020/21 winter and storms Christoph (RI=2.3) and Darcy (RI=1.6), with a maximum range
of 59 mm for $D_{50}$ April 2021, 98 mm for $D_{84}$ in August 2021 and 10 mm for $D_{16}$ in November 2021. At the end of the
monitoring period, all grain size metrics converged, most notably upstream of LD2 where $D_{84}$ coarsened to 5.5 mm
300 following winter 2021/22, while $D_{16}$ and $D_{50}$ downstream of the LD both fined to < 1 mm and 5 mm respectively from peaks
of 10.5 mm and 60 mm.



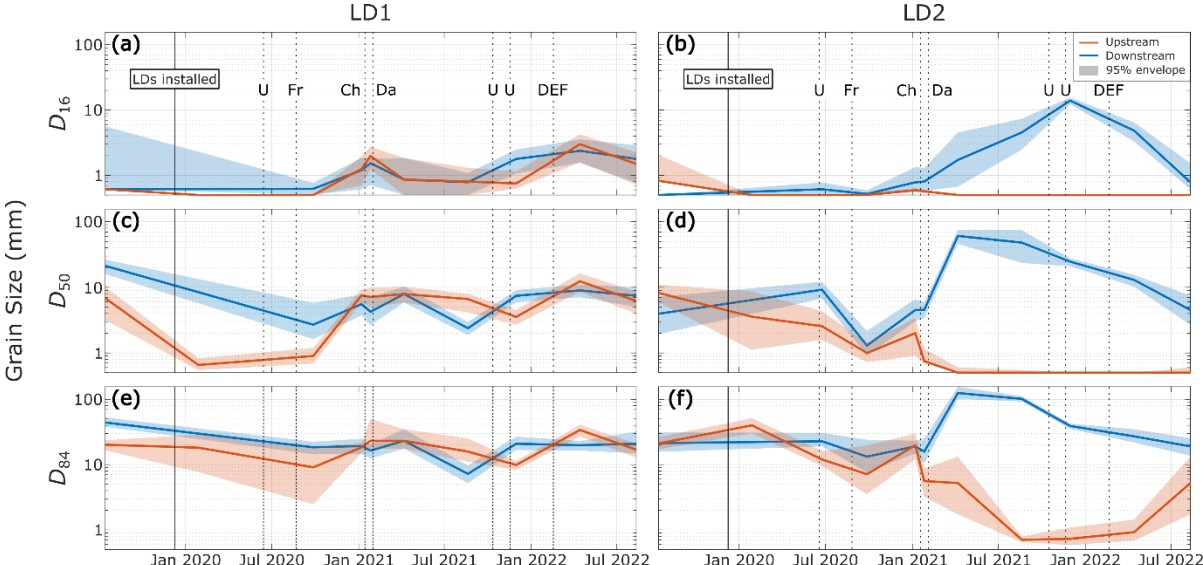

**Figure 4:** GSD metrics for upstream (orange) and downstream (blue) of the LD structures, with the 95% error envelope shown. Key storms are indicated by dashed lines.

## 3.3. Topographic variability

Topographic variability across each site was determined through first detrending all gridded data points in CloudCompare and computing the deviation of each point from the elevation mean. Surface elevations were shown to be normally distributed at both sites (Fig. 5a). LD1 had a minimum bed elevation range of 0.26 m in July 2019, before the installation of the LD, and a maximum of 0.36 m in January 2021. The deviation from elevation mean did not evolve over time and, although the elevation distribution fluctuated between surveys, the range and distribution of deviations remained similar throughout the monitoring period. Kurtosis was used to represent geomorphic variability and skewness the tendency for deposition or erosion. A distribution with a high-kurtosis represents a clustering around the mean (i.e., lower variability) and low-kurtosis represents more values in the tails of the distribution (higher variability). Skewness and kurtosis for LD1 was shown to vary between -0.2–0.8 and 1.4–2.4 respectively (Fig. 5b) with no clear temporal trend across the surveys, implying that the topographic variability remains unchanged throughout the monitoring period. There was also no generation or removal of existing bedforms, and bed elevation change was likely not influenced by the LD as further shown in Fig. 6.

In contrast, LD2 (Fig. 5c) exhibited increased topographic variability throughout time, with a minimum bed elevation range of 0.3 m in July 2019 and a maximum of 0.58 m in February 2022 following storms Dudley, Eunice and Franklin. The distribution evolved temporally from one that was predominately above the elevation mean with little variability (i.e., skewness > 1, kurtosis > 3.5; Fig. 5d) to a flatter, wider distribution with up to 0.38 m of deviation from the mean (February 2022) highlighting the development of plunge and underflow pools beneath LD2 (Fig. 7c1, d1). There was clear progression throughout time from the pre-LD survey in July 2019, stabilising following the wet winter in January 2021 and fluctuating




between survey dates. There was a clear relationship between skewness and kurtosis with time, with surveys becoming moderately positively skewed (0.5–1) with increasing geomorphic variability (decreasing kurtosis). Both skewness and

kurtosis fell between January and August 2021 when there were no substantial changes to stage—as shown in Fig. 2—and no storm events, but increased following storms Dudley, Eunice and Franklin towards winter 2022. The overall negative trend of the kurtosis indicates that the LD has increased internal topographic variability compared to the baseline observation, while fluctuating skewness generally identifies periods of accretion and erosion throughout the reach.

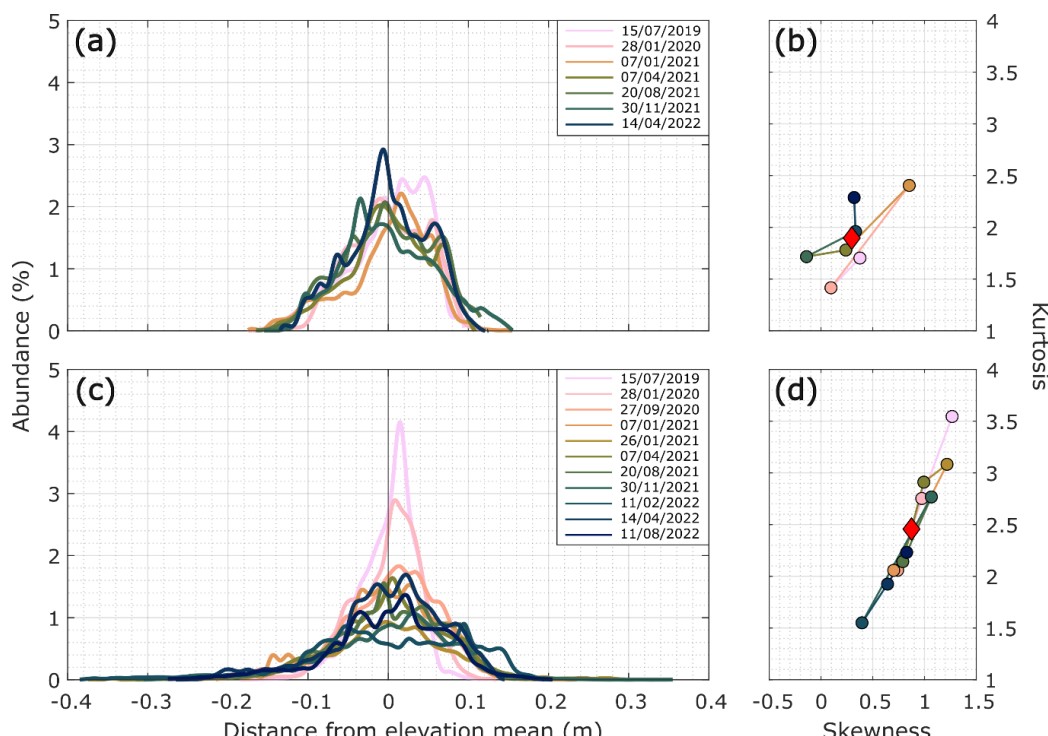


**Figure 5:** Detrended surface elevations relative to the mean for each bathymetric survey for LD1 (a) and LD2 (c). Associated skewness and kurtosis for each date is also shown with the mean shown as a red diamond (b) & (d) respectively.

### 3.4. Bed elevation change

Bed elevation was analysed through using TS data representing the 'wet' areas of the reach. LD1 had a maximum $U_{crit}$ of

0.0211 m (mean: 0.0180 m) while LD2 had a maximum of 0.0271 m (mean: 0.0184 m). Areas of change lower than the calculated LoD for each DoD are represented in grey in Fig. 6 and Fig. 7 for LD1 and LD2 respectively, however their abundance is preserved in each histogram. DoD histograms for both sites and surveys were normally distributed when preserving the stable areas.



LD1 exhibited a maximum scour depth of 0.16 m between January 2020 and November 2021 (Fig. 6b–e) which was
distributed throughout the reach, both upstream and downstream of the LD. There was a maximum deposition depth of 0.14
m between April and November 2021 (Fig. 6d, e), situated approximately 5 m downstream of the LD. The range of elevation
change did not fluctuate greater than ±0.04 m throughout time. Spatially, most of the change was situated greater than 5 m
from the LD, with the change around the LD being below the LoD threshold.

LD2, which was formed by two linked structures, had a maximum scour depth of 0.38 m between 26/01/2021 and April
2021 (Fig. 7e2), and a maximum deposition depth of 0.4 m (07/01/2021–26/01/2021; Fig. 7d2). The range of elevation
change fluctuated throughout time from -0.14–0.12 m immediately after LD installation (July 2019-January 2020; Fig. 7a2)
to reported maximum values. Additionally, the DoDs were highly spatially variable, with evidence of pool formation
downstream of the lower LD (Fig. 7a, b) which experiences periodic infilling (Fig. 7c, d) and scour (Fig. 7e, h) throughout
the monitoring period. Immediately downstream of the upstream LD, a secondary pool formed that exhibited similar scour
and fill cycles. The upstream LD stored sediment during the monitoring period, most clearly illustrated by Fig. 7d1
(07/01/2021–26/01/2021 following Storm Christoph), where a substantial area of the DoD exhibited change greater than the
LoD. The depositional zone also fluctuated over time, infilling and scouring on the true left bank of the upper LD.





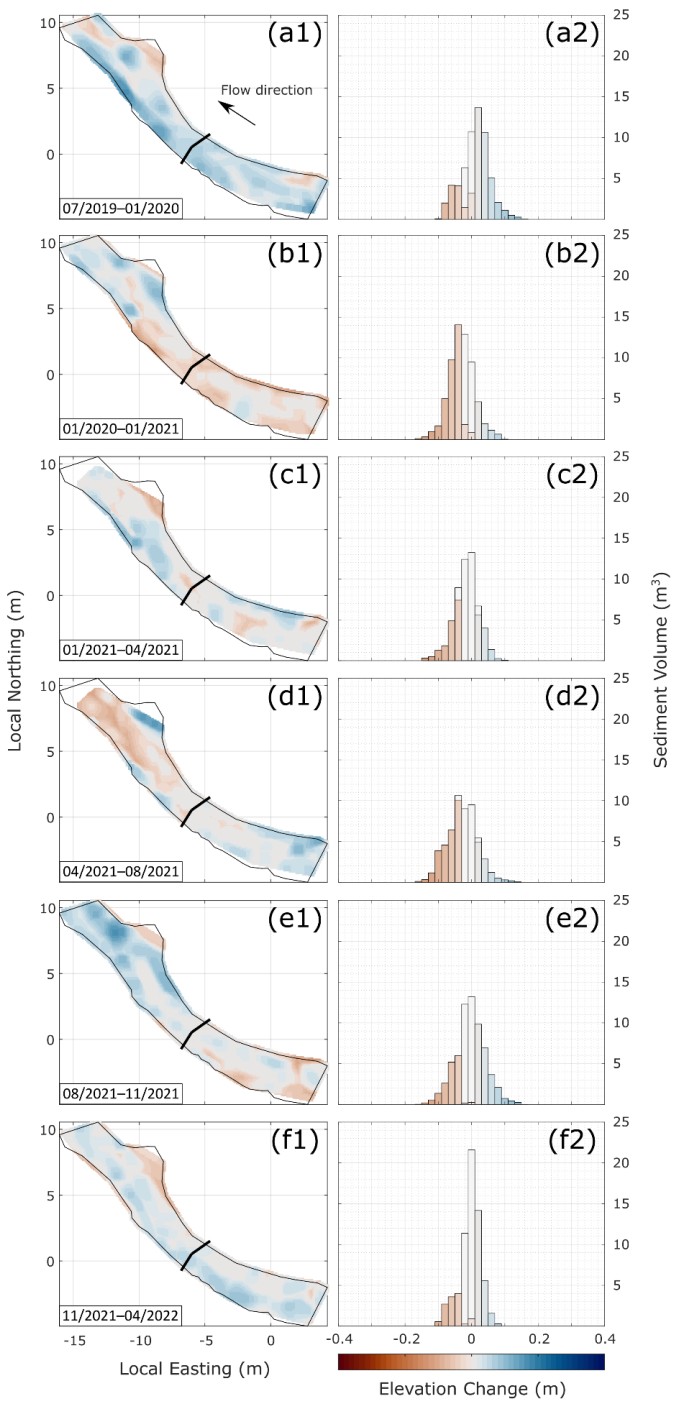

**Figure 6:** DoDs for LD1 (1) and histograms of elevation change (2). Significant elevation change ($a < 0.05$) denoted by colour (red is erosion, blue is deposition), with insignificant change shown in grey. The channel boundary is shown as a black outline, with the planar view of the LD represented as a thick black line.



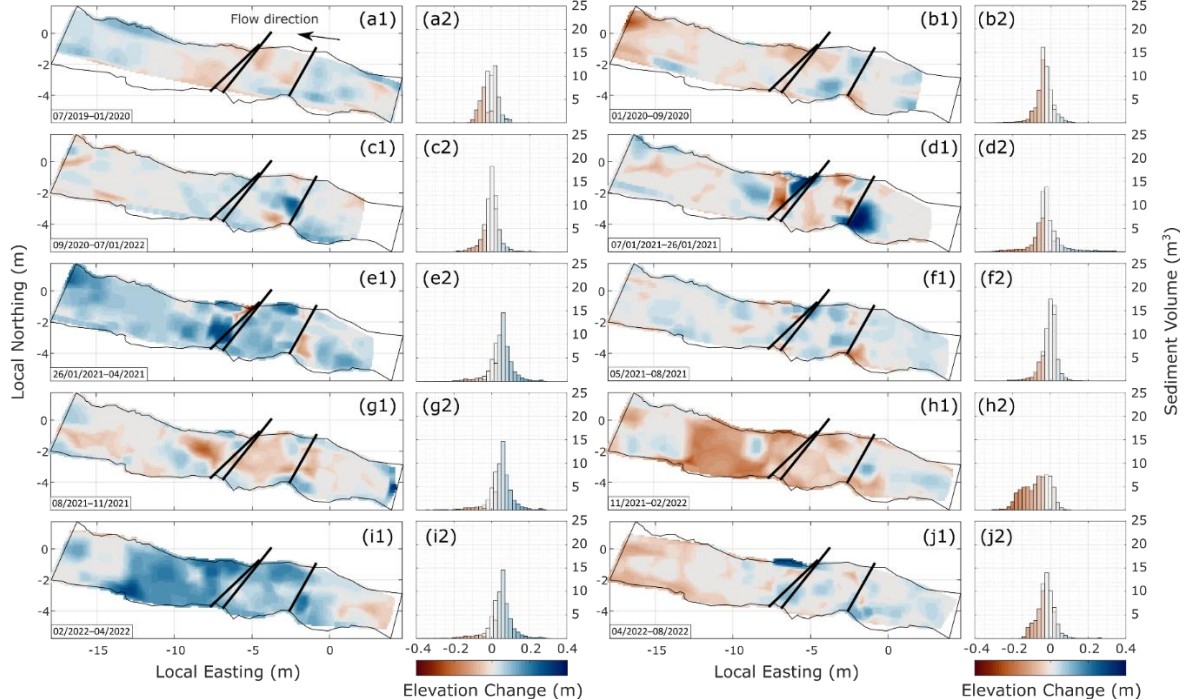

**Figure 7:** DoDs for LD2 (1) and histograms of elevation change (2). Significant elevation change ($a < 0.05$) denoted by colour (red is erosion, blue is deposition), with insignificant change shown in grey. The channel boundary is shown as a black outline, with the planar view of the LD represented as a thick black line. Flow direction denoted by black arrow.

Upstream of LD1 (Fig. 8a) volumetric change was highly variable, with no greater than 0.6 m³ of deposition immediately after LD installation and 1.1 m³ of erosion in January 2020. Upstream net change fluctuated throughout the monitoring period, with increased scour following higher flows in winter, and increased deposition across summer period lower flows. Downstream of LD1 (Fig. 8b) exhibited a similar pattern but there was increased erosion (maximum of 1.6 m³ in August 2021) and deposition (up to 1 m³ in November 2021).

Upstream of LD2 there was a clearer temporal trend (Fig. 8c). There was a maximum deposition of 0.6 m³ on 26/01/2021 and increased erosion (0.4 m³) following higher flows in February, with a clear progression as less sediment was being stored throughout the year. The LD impounded sediment upstream (Fig. 7d1) and scoured over the following year prior to infilling again. Although the range of volume change was smaller than upstream of LD1, there was clearly a progression linked to the presence of the LD. There was more variability mid-structure at LD2 ranging from a maximum of 1 m³ of erosion between January and February 2022 and 1 m³ of deposition between February and April 2022 (Fig. 8d), reflecting pool formation and cyclic scour and infilling between the two LDs (see also Fig. 7a1–d1). Downstream of the LDs (Fig. 8e) also exhibited similar variability, where plunge pools developed immediately downstream of the structure near the true right bank, followed by deposition further downstream. This section had the greatest range of volume change, with up to 3.4 m³ of erosion between February and April 2022, and 3.0 m³ of deposition between January and February 2022. Spatially, these





periods were less variable as there was reach-wide erosion and deposition, however there were localised areas 1 m downstream of the LD that exhibited little elevation change.

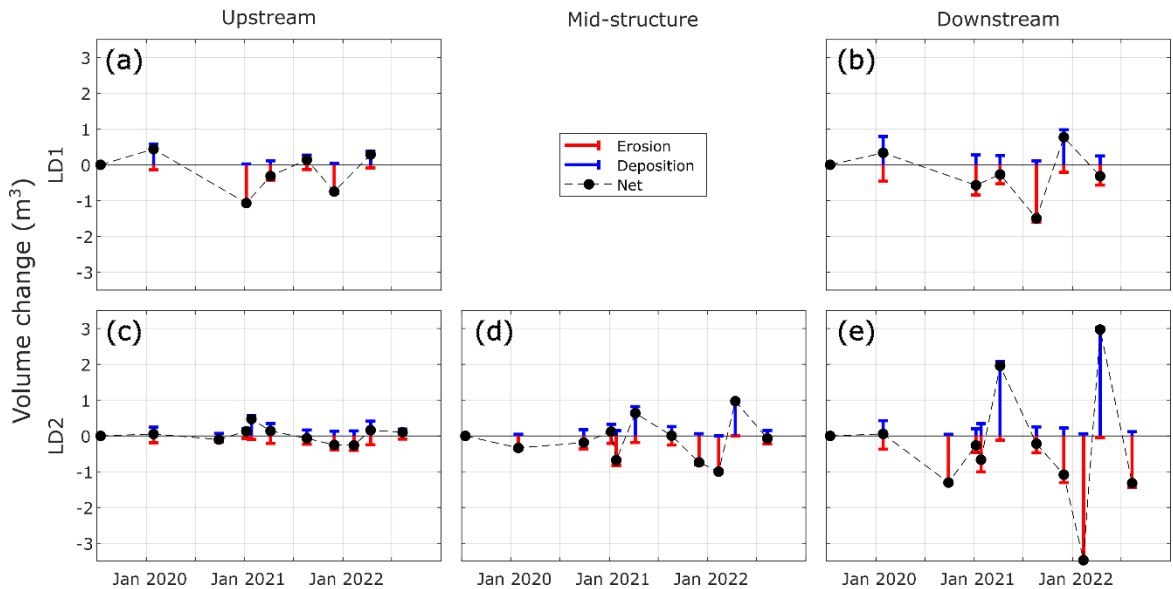

**Figure 8:** Volumetric change split into erosion and deposition for LD1 upstream (a) and downstream (b), and LD2 (c–e) separated into upstream of LD2b (c), mid-structure (i.e., between LD2a and LD2b; d) and downstream of LD2a (e).

### 3.5. Planform evolution

The banks at LD1 (Fig. 8a) predominantly exhibited erosion upstream of the LD and deposition downstream. The true right bank experienced 2.4 $m^2$ of bank erosion, and 4.8 $m^2$ of deposition, with a net change of 2.4 $m^2$ (Table 3). In contrast the true

left bank at LD1 predominantly exhibited erosion, experiencing -1.9 $m^2$ of bank loss, 0.41 $m^2$ of deposition and -1.5 $m^2$ of net change. Overall net planform area change at LD1 was 0.90 $m^2$. The channel at the site is relatively straight, and there was no clear influence from LD1 on bank erosion since the true right bank had a similar magnitude of planform change both above and below the LD.

LD2 (Fig. 9b) predominantly exhibited erosion, with a net planform area change of -1.10 $m^2$ but with greater spatial

variability near the LD sequence. Immediately downstream of the LD2b on the true right bank was an area of deposition, followed by approximately 0.50 $m^2$ of bank erosion, flanking LD2b. A similar pattern was observed on the true left bank, with an area of erosion upstream of LD2b, and a small area of deposition downstream. Often during high flows the river would exceed the river banks as highlighted above—and flank the true right bank of LD2b. This has clearly resulted in localised scour downstream of LD2b on the true right bank. LD2a appeared to have had little impact on the banks.



Deposition mainly occurred in the mid-structure, preceded and succeeded by bank erosion. There was less apparent impact from LD2a downstream, as shown in Fig. 9b.

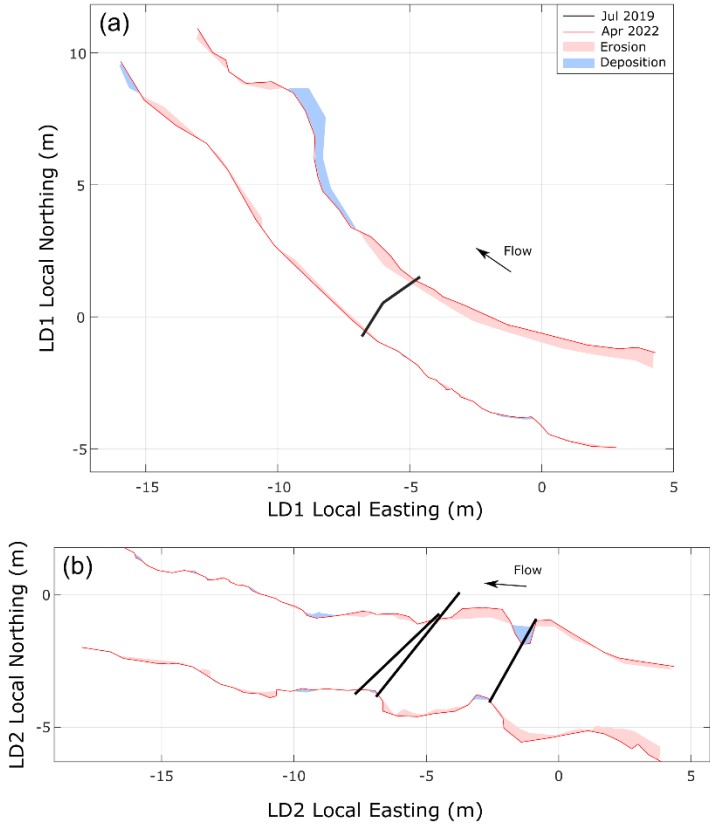

**Figure 9:** Bank erosion and deposition for LD1 (a) and LD2 (b). Leaky wooden dams are represented as thick black lines.

**Table 3:** Planform change for LD1 and LD2 between July 2019 and April 2022.

|  | True Bank | Erosion (m²) | Deposition (m²) | Bank net change (m²) | Site net change (m²) |
|---|---|---|---|---|---|
| LD1 | Right | -2.37 | 4.76 | 2.38 | 0.90 |
|  | Left | -1.90 | 0.42 | -1.49 |  |
| LD2 | Right | -0.74 | 2.21 | 1.48 | -1.10 |
|  | Left | -2.86 | 0.28 | -2.57 |  |

## 4. Discussion


This study has presented quantitative measurements of morphological change at two contrasting leaky wooden dam structures in the Dalby Forest, North York Moors, UK. It has shown that the presence of leaky wooden dams can substantially alter local topographic variability and influence GSDs if the structure is frequently engaged with the river flow and the sediment supply. This discussion aims to highlight the importance of monitoring multiple different facets of the river




reach, including monitoring consistently during periods of low flows, to elucidate the drivers for change. It was not possible to access a control reach in the study catchment, however data presented herein provides insights for long-term implications and suggestions for practitioners tasked with designing these structures are provided, whilst encouraging the continued collection of geomorphological data to build a database to inform future research.

### 4.1. Sedimentary response to varying flow conditions

Water depth for LD2 was regulated upon the activation of LD2b, increasing the depth upstream and decreasing downstream, most notably after the wet winter of 2020/21 where the study catchment experienced over 170% of winter rainfall compared to 1991–2020 (Kendon et al., 2022). LD2 was more effective at attenuating high flows and sustaining low flows during the monitoring period, and consistently maintained a difference between upstream and downstream water depth loggers. As such, the increased structural complexity of LD2 appears to be more effective at attenuating flows than LD1, aligning with

Cashman et al. (2021). In response to increased attenuation it is likely that the relative upstream flow velocity reduced in response to temporary blocking of the structure, allowing fines to settle out of suspension and deposit above the gravel riverbed increasing the proportion of fines upstream and an decreasing them downstream, similar to fully blocked natural wood jams (Welling et al., 2021; Wohl and Scott, 2017). Following this period few storm events occurred until October 2021, suggesting that low flows winnow finer sediments out of the downstream pool generated by overtopping of the leaky

wooden dam.

Large wood in rivers alters grain size and meso-scale bedforms (e.g., pools and riffles) often on the scale of 50 years or less (Montgomery et al., 2003), yet fluctuations in response to the hydrological regime and feedbacks with geomorphology highlight the importance of monitoring GSD throughout periods of both high and low flows. Sediment response is also controlled by sediment supply upstream, a reduction in sediment supply can increase the proportion of coarse sediment

downstream (e.g., Dietrich et al., 1989). Disconnectivity features (i.e., leaky wooden dams) can emulate and exacerbate this effect by impounding sediment upstream, thus reinforcing the armour bed in gravel bed rivers and limiting incision depth (Dietrich et al., 1989). LD2 replicates this behaviour when it becomes increasingly more blocked, effectively reducing sediment supply (especially coarser material) to zero, allowing the downstream bed to armour whilst enacting geomorphic change downstream, similar to naturally formed logjams (Cadol and Wohl, 2011). Though the impact of large dams for

hydropower are often reported as substantially altering the sediment supply (e.g., Bednarek, 2001; Kondolf, 1997; Piégay et al., 2019), LDs have the potential to cause disruption to the sediment supply when installed in a catchment system as a part of flood risk management, potentially resulting in unintended impacts downstream.

Equally important is the tendency of the GSD to reorganise towards being more similar both upstream and downstream of the structure, especially $D_{50}$ and $D_{84}$, however the monitoring period ends before they coalesce. This highlights the

importance of a study throughout time with frequent, detailed observations to ensure that variability in response to the flow regime is being correctly captured. The increased retention of sediments and ultimate flow regulation both up and





downstream induced through disconnectivity may result in increased stream power downstream of the leaky wooden dam intervention both conceptually as shown by Kondolf (1997), and in this study through the grain size disparity (Fig. 4). Check dams that were historically installed to regulate flow conditions are similar (albeit larger) to leaky wooden dams by creating

static points that induce disconnectivity have been shown to exacerbate incision downstream due to sediment starvation (Lo et al., 2022; Wohl and Beckman, 2014; Wyżga et al., 2021). There is therefore the potential risk that leaky wooden dams installed near to critical infrastructure may result in increased erosion (e.g., Nisbet et al., 2015) , therefore careful consideration of placement of these structures is required, especially in a wood-supply poor area. Additionally, if the structures do not provide an effective conduit between the active channel and the floodplain, there is potential for the

sediment stored to reduce the efficacy of the leaky wooden dam for flood risk management, however will likely provide water quality benefits (Quinn et al., 2013). Stored sediment will likely reduce the porosity of the structure (Lo et al., 2022), thereby further increasing longitudinal disconnectivity, similar to that of organic material (Schalko et al., 2018).

## 4.2. Influence on geomorphic complexity

The design of the leaky wooden dam is pivotal to forcing geomorphic work, and key to emulating and accelerating other

natural processes such as large wood recruitment, increased geomorphic heterogeneity, sediment storage and diverse habitat creation (Lo et al., 2021; Montgomery et al., 2003). Gap size was a key determinant for whether the leaky wooden dam was engaged by the flow or not for the two sites monitored herein. LD1 was largely ineffective at engaging with the channel across a host of conditions and consequently had little to no impact on instream hydraulics and morphology, with any change in grain size distribution, bed elevation and bank erosion likely within the range of natural variability.

The twin-structured LD2 was shown to have a more substantial impact on channel morphological response than LD1. There was a distinct evolution in the bathymetry of the reach up- and downstream of the leaky wooden dam, with increased deposition upstream and enhanced scour downstream. The leaky wooden dam actively influenced the morphology of the river channel through creating geomorphic heterogeneity shown through the evolution of the deviation from elevation means in Fig. 5c, creating important potential habitats for benthic macroinvertebrates as well as increased spatial variability of

benthic habitats in surrounding channels (Lo et al., 2021; Pilotto et al., 2016). Interestingly, the deviation from the mean quickly adjusted to the installation of the leaky wooden dam before becoming relatively stable between September 2020 and August 2021, with the largest period of adjustment between November 2021 and February 2022, likely as a result of Storm Barra (5–9 Dec 2021) where there was a total of 562 mm of precipitation averaged across the catchment (derived from the NIMROD radar system; Met Office, 2003. N.B. data for 08/12/21 are missing therefore true figure is likely higher).

Following this period, there was a period of accretion of 4 m3 (Fig. 8). It is clear from Fig. 7 that sediment was stored behind the upstream LD, but sediment also infilled the downstream pool, likely due to mobilised larger sediment from upstream, reflected in Fig. 4f where there is a decrease in all downstream grain size metrics. These observations align with the conceptual model of Faustini and Jones (2003) where large wood can detain finer sediment and form a semi-persistent



sediment wedge, as well as observations by others where fine sediment was retained by channel-blocking structures (Parker
et al., 2017; Welling et al., 2021), and bedload transport interrupted (e.g., Clark et al., 2022; Spreitzer et al., 2021). The
structure had an impact on the channel banks upstream of the sequence during high flows. The floodplain was partially
inundated and the leaky wooden dam flanked, creating new flow pathways, resulting in localised planform evolution
surrounding the leaky wooden dam, similar to natural wood. There was up to 0.5 m2 of localised bank retreat, and a similar
area of deposition immediately downstream of the LD. The change was confined to within 3 m upstream and downstream of
the individual LD, likely due to flow velocity reduction as a result of the backwater effect due to being engaged with the
flow (Schalko et al., 2023; Wohl and Beckman, 2014), however is a small amount. As the wood was relatively small in
comparison to the size of the channel, it may be prone to becoming dislodged during high flows in the future if it were not
anchored *in situ* (Dixon and Sear, 2014). However, the flanking may scour the anchor out, increasing the likelihood of leaky
wooden dam failure and exacerbating flood risk (Hankin et al., 2020).

**4.3. Long-term implications**

One of the key issues with natural flood management is scaling from the individual structure to the reach scale, and then to
the catchment scale, and understanding how different interventions can be implemented for the maximum overall benefit
(Ellis et al., 2021; Wingfield et al., 2021) or benefits to particular locations in the catchment for a given event. Many
structures are installed with a gap between the riverbed and the base of the leaky wooden dam (Lo et al., 2022; Wren et al.,
2022), however they are often not monitored due to limited funding and being regarded as a low-risk intervention. Despite
this, monitoring of leaky wooden dams is vital to understand their effectiveness, including pre- and immediately post-
installation topographic and grain size surveys. Quarterly monitoring supplemented with post-event surveys of the leaky
wooden dams studied herein captured extensive geomorphological change in response to seasonal variability and periods of
both high and low flows, including riverbed reorganisation that helped maintain flow depths during periods of low flows and
riverbank evolution within 1 m of the structures. To capture seasonal variability, it is recommended that structures are at
least checked following high flows or extended periods of low flows over a duration of at least 1 year. However, the
selection of an appropriate monitoring frequency and duration is challenging and will be dependent on the local climate,
hydrological and sedimentary regime. Identification of an optimal monitoring duration and frequency would help to
effectively conserve and distribute monitoring resources.

**5. Conclusions**

Leaky wooden dams are large wood structures installed into upland catchments to slow the flow of water, ultimately
reducing flood risk downstream, whilst emulating natural processes. This study aimed to quantify the geomorphologic and
sedimentary response to the installation of two leaky wooden dams installed in Dalby Forest, North Yorkshire, UK in
December 2019 using high-resolution spatial and temporal observations. Leaky wooden dams when engaged with the river

flow can enact substantial geomorphic change, increasing local topographic and sediment heterogeneity. This study found that grain size distributions can drastically vary throughout time up and downstream of leaky wooden dam structures when these structures are engaged with flows. Additionally, the results show how the local geomorphology and sediment distribution can organise back to a low flow state when sustained by consistent velocities from the leaky wooden dam structures. The work herein showed minimal impact on planform evolution at both monitoring sites, but this may be a

function of the leaky wooden dam designs and a study limitation of few flow events greater than RI=2 during the monitoring period.

Collection of high quality, high resolution topographic and grain size data are an important step towards addressing the structure-scale data paucity for LDs and the induced changes resulting from their installation, including sedimentary change, and understanding the magnitude and directionally of both geomorphic work and sediment grain size distribution evolution.

These are both important to collect throughout the duration of a project, and ideally following project completion.  This study highlights the importance of considering both the response of leaky wooden dams to high and low flows, and clearly illustrates the degree of topographic variability that can be induced by their installation.

## 6. Data availability

Data used in this study are available from https://doi.org/10.5281/zenodo.13832285, including grain-size distributions, water

depths, bathymetric elevations and bank profiles.

## 7. Author contributions

JW conceived the study design and performed the monitoring programme with support from CS and DM. JW performed the data analysis and writing of the first draft. RT performed the flow frequency analysis. JW, CS, DM, RT and DP equally contributed to discussing and interpreting the results and finalising the draft.

## 8. Competing interests

Daniel Parsons is a member of the editorial board of Earth Surface Dynamics.

## 9. Acknowledgements

The authors would like to acknowledge Dalby Forest and Forestry England for granting permission to perform the monitoring on their property. The authors would also like to than three previous anonymous reviewers of an earlier version

of this manuscript whose feedback improved the overall article.





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
