# Peer review of "Localised geomorphic response to channel-spanning leaky wooden dams"

_EGUsphere, 2024_

## Author Comment (AC1)

**Response to reviewers regarding Manuscript ID egusphere-2024-3001 entitled:**

**"Localised geomorphic response to channel-spanning leaky wooden dams" to Earth Surface Dynamics.**

We would like to thank Xiaofeng Liu and another anonymous Reviewer for their feedback on this manuscript, as well as Anastasia Piliouras for overviewing the submission and review process. We have responded to all Reviewer comments as detailed below.

Original text from the reviewers is blue.

Responses are black, Unedited text is grey, Edited text is **bold and black**.

**Reviewer 1 [Xiaofeng Liu]:**

This manuscript presents in great details the monitoring protocols, methods, and results at two field sites for installed leaky wooden dams. The data presented herein are potentially of great value to both fundamental river research and practice. Such data (and the in-depth analysis) are rare in the literature. The manuscript is also well organized and written, though some consolidation and reduction in length are needed. I recommend the publication of this work.

Thank you for the positive view of our work.

**Some of my specific comments are as follows:**

1. The Introduction section can be shortened. While comprehensive, there are many things are obvious, such as the lengthy description on the benefits of large wood and their impact.

Thank you for this comment. We have chosen to retain the introduction as it we believe it provides helpful links to further work related to leaky wooden dams, highlighting their interdisciplinary research interest.

2. Line 80: "A key unknown of is …". "of" should be removed.

Done.

3. The whole manuscript is full of technical details, such as software used, data processing algorithms, etc. It reads much like a project report, not a concise journal paper. I suggest move some of the nonessential details into supplementary materials.

Thank you for this comment. We have removed section 2.4.1. and 2.4.2. on Data Processing and Error Quantification of the topographic and bathymetric data to the supplementary information to reduce the length of the main manuscript.

4. Following on the comment above, I suggest the authors to add more analysis and explanations based on the data and observations. For example, Line 272 to 275: why at LD2 "Downstream peaks were generally shorter in duration than upstream"? This intrigues me. Don't the upstream and downstream water depths synchronize?

We have clarified the sentence (revised L241):

Downstream peaks were generally shorter in duration than upstream (Fig. 2b) **due to water being stored upstream thus extending the peak.**

5. Lines 285 onward for the GSD at LD2: I think what is observed makes sense. It is a dam after all. Fine materials deposit upstream in the pool and sediment coarsens downstream. A simple sentence like this makes the manuscript reads better rather than description only.

Thank you, we agree with your comment and have updated the text.

6. The authors need to comment on the limitation of their bed elevation analysis. They assumed the change of bathymetry is 100% due to the installation of LDs. Could be something else, e.g., upstream changes and fluctuations of sediment supply? Even without LDs installed, could there be any natural temporal variations of bathymetry over time?

It is possible that there were fluctuations upstream in terms of sediment supply, however through LD1 not activating throughout the duration of the monitoring period it has acted as a control study and captured the natural variability of the channel. This is now more explicitly mentioned in Section 4.2. (L420).

7. Did the LDs change over time? For example any recruitment of debris, clogging, or change of LDs itself due to decay, abrasion, and other processes?

The LDs did change over time. Wood and leaf debris were accumulated and lost between each visit but we are unsure how these changed between field visits. These are interesting topics for future research but are beyond the current scope of this paper.

**Reviewer 2:**

The objective of this paper is to evaluate the influence of two different LD (Leaky Dam) structures on the directionality and magnitude of geomorphic changes, geomorphic variability induced by the structures, and the importance of frequent monitoring to accurately assess long-term impacts at the local, unit scale. The study quantifies flow, geomorphic, and sedimentary impacts of two wooden LDs, one consisting of a single log and the other made of two sequential logs, over a 2.5-year period. Results include water depth, grain size distribution (GSD), and local morphology changes.

Overall, the paper is well-structured and provides valuable long-term field data and detailed statistical analysis, which is rare in real-world leaky dam studies. It holds significant scientific merit and quality. I recommend following minor revisions to further strengthen the manuscript.

Thank you for your positive review of our work.

**Specific Comments:**

1. Line 121: Please consider adding the minimum, mean, and maximum gap sizes for LD1, LD2a, and LD2b for easier comparison.

Mean gap is already presented in the text that is reflective of each LD as they were relatively parallel to the bed when installed. To make this clearer, we have stripped the key values from the text and introduced a new table (Table 1) to highlight each LD's length, vertical gap (mean/min/max) and extension onto the floodplain.

2. General results comment: When reporting parameter changes such as water depth, it would be beneficial to briefly attribute them to the physical mechanisms responsible.

Thank you for this comment. We attribute the changes to each parameter throughout the Discussion.

3. Figure 4: Why are upstream and downstream D16, D50, and D84 values similar before January 2021 but diverge afterward? A discussion of the mechanisms driving changes in grain size metrics over time would enhance understanding.

Thank you for this observation. We introduce the grain size metrics in Section 3.2 (L266) and discuss this divergence in detail in Section 4.1. The similarity in upstream and downstream grain size metrics (D16, D50, and D84) before January 2021 reflects the initial equilibrium condition of the system prior to significant hydrological events. The divergence after January 2021 coincides with a series of high-magnitude storms that occurred between January-March 2021, which triggered selective transport of finer sediments through the LDs. This selective transport resulted in coarsening downstream as finer fractions were preferentially mobilized while larger particles remained trapped upstream.

4. Line 293: Could you clarify the relationship between flooding events and bed response, specifically how it compares in cases with and without interacting logs?

We discuss this in Section 4.2 and have explicitly noted that LD1 was not activated therefore acts as a control site for the study period, as noted by Reviewer 1.

5. Line 395: Clarification on the LD2 Structure Effectiveness. Is the observed increased flow complexity and attenuation at LD2 primarily due to LD2b or LD2a.

LD2a had increased complexity and stored more water upstream owing to increased material blocking the channel (and then increasing sediment storage). LD2a also had increased

complexity owing to not being a single channel spanning log and interacted more with flow, capturing material. However, it did not have the same impact on water storage (and sedimentation) as LD2b, as represented in Figure 7.

We have updated the section as follows to reflect this (L364–L367):

> LD2a appeared to have had little impact on the banks**, despite retaining water**. Deposition mainly occurred in the mid-structure, preceded and succeeded by bank erosion. There was less apparent impact from LD2a downstream, as shown in Fig. 9b **as despite having increased structural complexity, the structure engaged with deeper flows less frequently than LD2b.**

6. Line 410 states, "Water depth for LD2 was regulated upon the activation of LD2b." Could you clarify how much LD2a contributed to flow engagement? If LD2a has a minor role, what explains the significant differences between LD1 and LD2b, given they are the same log type?

LD1 and LD2b are the same log type but LD2b captured more material and engaged with the flow more regularly, therefore having more impact. This led to the conclusion regarding gap size being more important (and supporting others in the literature). To clarify this further, we have updated the text in Section 4.1. as below (now lines L381–L387) to address this comment:

> Water depth for LD2 was regulated upon the activation of LD2b**, and minorly by LD2a,** increasing the depth upstream and decreasing downstream, most notably after the wet winter of 2020/21 where the study catchment experienced over 170% of winter rainfall compared to 1991–2020 (Kendon et al., 2022). LD2 was more effective at attenuating high flows and sustaining low flows during the monitoring period, and consistently maintained a difference between upstream and downstream water depth loggers. As such, the increased structural complexity of LD2 **(two LDs in sequence that captured more woody material than LD1)** appears to be more effective at attenuating flows than LD1, aligning with Cashman et al. (2021).

7. Lines 414 & 451: Line 414 discusses LD2's effectiveness, while Line 451 highlights gap size as key for flow engagement. Could you clarify if LD2's greater effectiveness stems from its structural complexity or if it's simply because LD1 did not interact with the flow due to the gap beneath it?

This comment is addressed partially in the response above. Gap size played a large role but since LD2b was higher than LD1 (mean distance from bed to base of the LD was 0.41 m vs 0.28 m, respectively) it recruited more woody material, enabling it to engage with the flow more frequently than the relatively simple structure of LD1.

8. Porosity/Blockage Ratio: Porosity is briefly mentioned. Could you specify the porosity or blockage ratio of the LDs and provide a comparison? Did these values change over the study period?

We did not calculate the porosity/blockage ratio of the structure as this was beyond the scope of this study.

9. Design Implications: What recommendations can you offer practitioners regarding LD design based on the site specifics such as in straight versus more complex channels?

We do not believe that our findings can offer these recommendations due to the study area and the lack of engagement from LD1, as addressed above. Our findings primarily relate to structural aspects of log dams themselves (such as gap size and its impact on grain size and bathymetric change) rather than providing guidance on placement within different channel configurations. However, we believe that this would be an interesting and useful inclusion for future study, especially when integrated with numerical modelling to systematically address the question.

**Minor Editorial Suggestions:**

1. Project Information: Including cross-sectional profiles and channel characteristics at the site would be helpful. A schematic diagram showing the channel layout and log positions would enhance clarity.

The location of the leaky wooden dams and channel banks can be seen in Figure 7, therefore we have not added the schematic figure to ensure the manuscript is concise.

2. Figure 1(c): Consider adding an arrow indicating flow direction.

Added.

3. Line 261: Specify the location where increased scour occurs.

Updated:

The true right bank became partially inundated and flow outflanked LD2b, resulting in increased scour **immediately downstream of LD2b**, limiting the maximum capacity of the structure during these bankfull periods.

4. Line 283: The reference to "Fig. 8a" needs to be changed to "Fig. 9a." Please verify and correct.

We believe this comment is in refence to line 383 and have updated to Fig. 9a accordingly.

5. Figure 9: Are changes relative to July 2019? July 2019 line is shown in the legend, but it is not visible in the figure.

Yes, we have updated the figure to ensure that the black line that represents July 2019 is there.

---

## Author Response (AR2)

**Response to reviewers regarding Manuscript ID egusphere-2024-3001 entitled:**

**"Localised geomorphic response to channel-spanning leaky wooden dams" to Earth Surface Dynamics.**

We would like to thank Xiaofeng Liu and another anonymous Reviewer for their feedback on this manuscript, as well as Anastasia Piliouras for overviewing the submission and review process, and Wolfgang Schwanghart for accepting the article.

We have corrected all Figure issues to ensure that the formatting remains consistent throughout the manuscript.

We have, however, left the river channel as yellow in Fig. 1 as the line style used for the channel is proportional to stream order rather than representative of the land cover map classification, and this colour enhances readability of the figure.

On behalf of my co-authors, we thank you for the time and effort by all in reviewing and editing the manuscript and look forward to the final publication in ESurf.

Kind regards,

Josh Wolstenholme